# Nationwide Newborn Screening Program for Mucopolysaccharidoses in Taiwan and an Update of the “Gold Standard” Criteria Required to Make a Confirmatory Diagnosis

**DOI:** 10.3390/diagnostics11091583

**Published:** 2021-08-31

**Authors:** Chih-Kuang Chuang, Chung-Lin Lee, Ru-Yi Tu, Yun-Ting Lo, Fran Sisca, Ya-Hui Chang, Mei-Ying Liu, Hsin-Yun Liu, Hsiao-Jan Chen, Shu-Min Kao, Li-Yun Wang, Huey-Jane Ho, Hsiang-Yu Lin, Shuan-Pei Lin

**Affiliations:** 1Division of Genetics and Metabolism, Department of Medical Research, MacKay Memorial Hospital, New Taipei City 25160, Taiwan; mmhcck@gmail.com (C.-K.C.); clampcage@gmail.com (C.-L.L.); likemaruko@hotmail.com (R.-Y.T.); fransiscazen@gmail.com (F.S.); 2College of Medicine, Fu-Jen Catholic University, Taipei 24205, Taiwan; 3Department of Pediatrics, MacKay Memorial Hospital, Taipei 10449, Taiwan; 4The Rare Disease Center, MacKay Memorial Hospital, Taipei 10449, Taiwan; andy11tw.e347@mmh.org.tw (Y.-T.L.); wish1001026@gmail.com (Y.-H.C.); 5The Chinese Foundation of Health, Neonatal Screening Center, Taipei 10699, Taiwan; meiying@cfoh.tw (M.-Y.L.); hyliu@cfoh.tw (H.-Y.L.); hjchen@cfoh.tw (H.-J.C.); kao@cfoh.tw (S.-M.K.); 6Taipei Institute of Pathology, Neonatal Screening Center, Taipei 10699, Taiwan; yurilarc@gmail.com (L.-Y.W.); juju5113@gmail.com (H.-J.H.); 7Department of Early Childhood Care and Education, Mackay Junior College of Medicine, Nursing and Management, Taipei 11260, Taiwan; 8Department of Medicine, MacKay Medical College, New Taipei City 25245, Taiwan; 9Department of Medical Research, China Medical University Hospital, China Medical University, Taichung 40402, Taiwan; 10Department of Infant and Child Care, National Taipei University of Nursing and Health Sciences, Taipei 11219, Taiwan

**Keywords:** mucopolysaccharidosis, lysosomal storage disease, newborn screening for MPS, gold standard of MPS confirmation, GAG-derived disaccharide

## Abstract

Mucopolysaccharidoses (MPSs) are a group of lysosomal storage diseases (LSDs) caused by an inherited gene defect. MPS patients can remain undetected unless the initial signs or symptoms have been identified. Newborn screening (NBS) programs for MPSs have been implemented in Taiwan since 2015, and more than 48.5% of confirmed cases of MPS have since been referred from these NBS programs. The purpose of this study was to report the current status of NBS for MPSs in Taiwan and update the gold standard criteria required to make a confirmative diagnosis of MPS, which requires the presence of the following three laboratory findings: (1) elevation of individual urinary glycosaminoglycan (GAG)-derived disaccharides detected by MS/MS-based assay; (2) deficient activity of a particular leukocyte enzyme by fluorometric assay; and (3) verification of heterogeneous or homogeneous variants by Sanger sequencing or next generation sequencing. Up to 30 April 2021, 599,962 newborn babies have been screened through the NBS programs for MPS type I, II, VI, and IVA, and a total of 255 infants have been referred to MacKay Memorial Hospital for a confirmatory diagnosis. Of these infants, four cases were confirmed to have MPS I, nine cases MPS II, and three cases MPS IVA, with prevalence rates of 0.67, 2.92, and 4.13 per 100,000 live births, respectively. Intensive long-term regular physical and laboratory examinations for asymptomatic infants with confirmed MPS or with highly suspected MPS can enhance the ability to administer ERT in a timely fashion.

## 1. Introduction

Mucopolysaccharidoses (MPSs) are a group of lysosomal storage diseases caused by mutations of genes encoding for lysosomal enzymes leading to defects in the stepwise degradation of glycosaminoglycans (GAGs). The accumulation of GAGs in lysosomes can result in cellular dysfunction and clinical abnormalities. Clinical manifestations of MPS are chronic and progressive. MPS patients present with many distinct clinical features including organomegaly, developmental delay, dysmorphic facial features, skeletal dysplasia (dysostosis multiplex), and growth retardation. In addition, hearing, vision (mostly with corneal clouding), cardiovascular function, and joint mobility are also affected. The clinical features of MPS may present from birth to late childhood or even early adulthood depending on the severity of the MPS phenotype. The onset of disease is commonly between the ages of 18 months and 4 years. Death from respiratory or cardiac failure and respiratory infections usually occurs before the age of 10 years in patients with severe phenotypes [1,2,3,4]. Except for MPS II, which is X-linked, all MPSs are autosomal recessive disorders. Pilot newborn screening (NBS) programs for MPS I, II, IVA, and VI have been progressively implemented in Taiwan since 2015, and 48% (16/33) of patients diagnosed with MPS between 2016 and 2019 were referred from these NBS programs. The median age at diagnosis of these newborns was 0.2 years, which is notably lower than the median age of 4.3 years in the other 159 patients who were diagnosed due to clinical indications [5].

As of 30 April 2021, more than 600,000 infants enrolled in this program have been distinguished for MPS type I, II, VI, and IVA, including 599,962 for MPS I, 307,731 for MPS II, 72,490 for MPS IVA, and 447,389 for MPS VI. Individual MPS NBS programs have been implemented since August 2015. A total of 255 infants have been referred to MacKay Memorial Hospital for a confirmatory diagnosis. Among these infants, four were confirmed to have MPS I, nine MPS II, and three MPS IVA, with prevalence rates of 0.67, 2.92, and 4.13 per 100,000 live births, respectively. These findings show little change compared to data reported in 2018 for MPS I (1.36/100,000 live births) and MPS II (1.96/100,000 live births) [6]. These 234 infants were defined as being “positive” on the basis of lower enzyme activity in dried blood spot (DBS) analysis by tandem mass spectrometry assay. Increasing laboratory evidence has shown that the conventional “gold standard” criteria used to confirm MPS, i.e., deficient activity of a specific leukocyte enzyme, is no longer adequate to make a confirmatory diagnosis and can lead to a false positive diagnosis or misdiagnosis of MPS. Leukocyte enzyme activity is affected by the pathogenicity of nucleotide variations found in the gene causing MPS, regardless of whether the clinical manifestations are severe or mild. GAG-derived disaccharides including dermatan sulfate (DS), heparan sulfate (HS), and keratin sulfate (KS) as measured by tandem mass spectrometry assay have been shown to be effective biomarkers for MPS [7,8,9]. The gross accumulation of GAG-derived disaccharides can be identified in urine, blood, soft tissue, skeletal system, and even various organs from fetus to childhood. The definition of a true “positive” confirmatory diagnosis of MPS should be redefined as a “positive” urinary biochemistry examination, particularly an elevated quantity of urinary (u)GAG-derived disaccharides, deficient activity (less than 5% of normal activity) of a particular leukocyte enzyme, and verification of heterogeneous or homogeneous variants by Sanger sequencing or next generation sequencing.

Genotyping by Sanger sequencing can provide information on nucleotide variations. However, whether the variants are pathogenic or non-pathogenic, and whether this will affect the expression of enzyme activity needs to be assessed further. The impaired degradation of GAGs catalyzed by the dysfunction of enzyme activity will result in the gross accumulation of GAGs in cells, tissue and organs. In particular, the amount of GAG-derived disaccharides excreted in urine plays a crucial role in confirming MPS. Therefore, the aim of this study was to assess and report the relationships among the findings of genotyping, the expression of leukocyte enzyme activity, and the amount of GAG-derived disaccharides excreted in urine in order to make an accurate diagnosis of MPS.

## 2. Materials and Methods

### 2.1. NBS for MPSs and Samples Referred

The purpose of the NBS for MPS is mainly to detect the specific enzyme activities in DBS that is used to screen out the positivity of having MPS in newborns. Enzyme activities can be measured by using tandem mass spectrometry-based assay that can allow simultaneous measurement of multiple enzyme activities, including MPS I, II, VI, and IVA. All the substrates were commercially purchased from PerkinElmer, Inc. (Waltham, MA, USA). The protocol of the multiplex enzyme assay is described as followings. A 3-mm punch of DBS was incubated in buffer containing four substrates and internal standards overnight. A liquid-liquid extraction by using aqueous NaCl and ethyl acetate was performed. Subsequently, the ethyl acetate layer was then collected and dried. The residue was consequentially resuspended in solvent for auto-sampling for tandem mass spectrometry analysis. All of the 255 infants with suspected MPS (12 MPS I, 186 MPS II, 42 MPS IVA, and 15 MPS VI) were referred to MacKay Memorial Hospital (MMH), Taipei, Taiwan, for further confirmation, including regular physical examinations, urinary first-line biochemistry examinations, leukocyte enzyme assay, and molecular DNA analysis at the first visit to the outpatient department. Sixteen of the 255 suspected infants who were referred from the NBSC (Newborn Screening Center) were confirmed to have MPSs, including four with MPS I, nine with MPS II, and three with MPS IVA, all with unrelated families except for the infants with MPS I who were twin sisters and siblings of a brother and sister. The other 239 infants were defined as being “highly suspected of having MPS”, “MPS carriers or having pseudo MPS” and “not having MPS” according to the results obtained from biochemistry tests and molecular DNA analysis. Urine (10–20 mL) and EDTA blood (2 tubes, 3–5 mL in each) samples were collected for the confirmatory diagnosis. The urine samples were stored at −20 °C prior to GAG analyses, and the blood samples were kept at room temperature and 4 °C before leukocyte isolation for enzyme assay and molecular DNA analysis, respectively.

### 2.2. Ethical Approval

All experimental protocols were approved by the Institutional Review Board of MMH (20MMHIS450e and 20CT022be), and the methods were carried out in accordance with relevant guidelines and regulations. Informed consent was obtained from all participants and/or their legal guardian/s.

### 2.3. Urinary First-Line Biochemistry Examinations

The urinary first-line biochemistry examinations included two well-established tests to quantity the total amounts of GAGs and individual GAG-derived disaccharides.

#### 2.3.1. Total GAG Quantification (Dimethylene Blue/Creatinine Ratio; DMB/Cre Ratio)

The DMB/Cre ratio method has been described previously [6,10]. GAGs are determined quantitatively in urine by reaction with the dye dimethylmethylene blue (DMB) in a reaction that does not require prior precipitation of the GAGs. The color is measured rapidly at wavelength 520 nm. The DMB ratio is obtained by dividing the urine creatinine with GAG volume in mg/L, and the ratio is expressed as mg/mmol creatinine. The DMB/Cre ratio gives an estimation of the GAG concentration in urine. It is age-dependent, and the lower the age, the higher the DMB/Cre ratio. The DMB/Cre ratio can be used as a reference diagnosis for MPS, but it cannot be used to determine the type of MPS. The normal reference values based on age group are, <6 months: <70.68 mg/mmol Cre (mean ± 2SD = 41.83 ± 28.85); 6 months-2 years: <46.80 mg/mmol Cre (mean ± 2SD = 22.28 ± 24.52); 2-17 years: <20.98 mg/mmol Cre (mean ± 2SD = 10.99 ± 9.99); and >18 years: <12.62 mg/mmol Cre (mean ± 2SD = 4.23 ± 8.39).

#### 2.3.2. GAG-Derived Disaccharide Quantification by Tandem Mass Spectrometry Assay

The use of tandem mass spectrometry assays to quantify GAG-derived disaccharides has been reported previously [11,12,13]. In this study, a chemical hydrolysis and an enzyme digestion method were applied to quantify CS, DS, HS, and KS, respectively. Methanolysis was performed by adding 3N HCl in methanol. The methyl group (CH_3_) would bind to the COO^−^ (carboxyl group) of C6 of l-iduronate and to the C1 negative oxide ion of N-acetyl-galactosamine-4-sulfate. One particular disaccharide for each GAG was selected. The treatment of keratanase II extracted from *Bacillus* sp., which cleaves *N*-acetylglucosamine linkages of the KS chain, releasing Galβ1-4GlcNAc (N-acetylglucosamine) disaccharides with mono-sulfates. In the multiple reaction monitoring (MRM) mode, the mass spectrometer detected ions by monitoring the decay of the *m/z* (mass-to-charge) ratios of the parent ion and its daughter ion after collision are 426.1→236.2 for DS, 384.2→161.9 for HS, and 462.0→97.0 for KS. Using normalization to urinary CS (in μg/mL) instead of the normalization to μg/mg creatinine can prevent false positives and false negatives for DS, HS, and KS quantification [14]. The sensitivity (true positive rate), specificity (true negative rate), and positive predictive values using the CS-normalized method for calculations were all 100%. Use of the MS/MS-based method for the quantitative analysis of GAG-derived disaccharides in physiological fluids has been shown to increase the accuracy of an MPS diagnosis. The cut-off values for DS, HS, and KS were <0.80 μg/mL, <0.78 μg/mL, and <7.90 μg/mL, respectively.

### 2.4. Leukocyte Enzyme Activity by Fluorometric Assay

All the 4-methylumbelliferyl substrates for enzyme fluorometric assay including IDUA, IDS, and GALNS were purchased from Toronto Research Chemicals Inc. (Toronto, ON, Canada), and the substrate, 4-nitrocatechol, used for ARSB enzyme spectrophotometric assay was purchased from Sigma-Aldrich Inc. (St. Louis, MO, USA). The protocols of enzyme assays have been described previously [6,15,16,17]. Enzyme activity was proportional to the amount of liberated fluorescence detected (μmol enzyme activity/g protein/h). The reference ranges for leukocyte IDUA, IDS, GALNS, and ARSB were 4.87–54.71 μmol/g protein/h, 12.89–131.83 μmol/g protein/4 h, 5.9–27.8 μmol/g protein/h, and 14–228 μmol/g protein/h, respectively. Individual enzyme activity about 5% lower than normal was defined as a marked reduction (deficiency) in that enzyme activity, and this is the diagnostic basis (gold standard) to confirm MPS.

### 2.5. Molecular DNA Analysis

Genomic DNA was extracted from peripheral blood leukocytes by high-salt extraction. Polymerase chain reactions (PCRs) of exons found in individual MPS types comprising adjacent intronic regions were performed with various designed primers and appropriate conditions. PCR amplification of cDNA or genomic DNA in patients and unaffected controls was carried out using oligonucleotide primers, i.e., IDUA (NG_008103.1) exon 1–14, IDS (NG_011900.2) exon 1–9, GALNS (NG_008667.1) exon 1–14, and ARSB (NG_007089.1) exon 1–8. PCR products were purified and sequenced using a DNA sequencer [18]. All amplified fragments flanking the exons were analyzed to identify variations. The resultant sequences were imported into Sequence Navigator software (Sequence Scanner Software 2, Applied Biosystems Inc., Foster City, CA. USA) for alignment, editing, and mutation analysis.

### 2.6. Verification of a New “Gold Standard” for MPS Confirmation

To confirm MPS, the “gold standard” is conventionally defined as the deficiency or functional loss of the activity of a particular enzyme that catalyzes the step-wise degradation of GAGs. In this study, we investigated the relationships among the findings of variant genotyping, leukocyte enzyme activity expression, and uGAG-derived disaccharide accumulation for individual cases referred from the NBSC. The infants were classified into four MPS diagnostic groups according to these findings (Figure 1). For the first group (the confirmed MPS group), the result categories were defined as “positive” for uGAG biochemistry examinations, “deficiency” for the activity of a particular enzyme, and the identification of one hemizygous variant for MPS II (x-linked recessive inheritance), or either two homozygous or heterozygous variants for MPS I, IVA, and VI (autosomal recessive inheritance). For the second group (the highly suspected MPS group), the categories were “negative” for uGAG biochemistry examinations, “deficiency or reduction” for the activity of a particular enzyme, and the identification of more than one known or novel gene variant. For the third group (the carrier or pseudo-MPS patient group), the categories were “negative” for uGAG biochemistry examinations, “reduction” for the activity of a particular enzyme, and the identification of only one known or one novel variant inherited from either parent. For the fourth group (non-MPS group), the categories were “negative” for uGAG biochemistry examinations, “normal” for the activity of a particular enzyme, and no gene mutations being found.

## 3. Results

Four MPS diagnostic groups were classified according to the results obtained from urinary and blood biochemistry examinations and molecular DNA analysis as follows: MPS confirmed group, MPS highly suspected group, MPS carrier or pseudo-MPS group, and non-MPS group (Figure 1). The urinary first-line biochemistry examination results including the DMB/Cre ratio and quantification of uGAG-derived disaccharides according to the type of MPS are shown in Table 1, Table 2, Table 3 and Table 4 [18,19,20,21,22,23,24,25,26,27,28,29,30,31,32,33,34,35,36,37,38,39,40,41,42,43,44,45]. A “positive” uGAG test was defined as a value of DS, HS or KS in urine significantly higher than the cut-off value, i.e., 0.80, 0.78, and 7.90 μg/mL, respectively. A higher DMB/Cre ratio than the normal reference value was not required to confirm MPS; however, some patients with MPS showed a remarkable increase. In addition to the uGAG test results, the nucleotide variation(s), protein variation(s), and enzyme activity are also shown in the tables. There were 13 mutations of the *IDUA* gene, 21 mutations of the *IDS* gene, 21 mutations of the *GALNS* gene, and 15 mutations of the *ARSB* gene underlying Taiwanese suspected MPS infants (Figure 2, Figure 3, Figure 4 and Figure 5). Comparisons of the results were performed in order to identify the best biomarker to confirm a diagnosis of MPS.

### 3.1. Interpretation of the Results for MPS I

Twelve infants were referred from the NBSC for MPS I confirmation. The IDUA activities in DBS were all lower than the cut-off values in the first test (<3.0 μmol/L/h) and second test (<1.5 μmol/L/h) [38]. Of these 12 infants, four were confirmed to have MPS I, including a brother and sister (infant No. 8 and 9) and twin sisters (infant No. 10 and 11), in whom missense nucleotide variations, i.e., [c.300-3C>G]+[c.1874A>C] and [c.1037T>G]+[c.1091C>T], caused the deficiency of IDUA enzyme activity. In addition, the uGAG tests were defined as being positive with significant elevations of DS and HS. Five infants (infant No. 1, 2, 3, 7, and 12) were classified into the highly suspected MPS group. These infants all carried two mutations, i.e., [c.2T>C]+[c.343G>A], [c.1081G>A]+[1395delC], [c.355G>T]+[c.617C>T], [c.76G>A]+[c.911delT], and [c.1093C>G]+[c.1463G>C]+[c.1828+5G>A], respectively, which caused the reduction in IDUA enzyme activities from 0.95 to 21.60 μmol/g protein/h; however, the uGAG tests were defined as being negative, and the amounts of DS and HS detected by MS/MS-based assay were all lower than the cut-off values. One carrier (infant No. 4) was verified to have a nucleotide variation, i.e., [c.179A>C], that caused the reduction in IDUA enzyme activity. This infant had showed normal amounts (<cut-off values) of DS and HS in urine. Finally, two infants (infant No. 5 and 6) were classified into the non-MPS group, in whom no *IDUA* gene variations were found. The IDUA enzyme activities were normal in one infant and slightly reduced in the other; however, they were normal in the follow-up examination (6 months later). The uGAG tests for these two infants were negative.

### 3.2. Interpretation of the Results for MPS II

A total of 186 infants were referred from the NBSC to MMH for MPS II confirmation, and their IDS enzyme activities were all lower than the cut-off values in both the first and second tests (<3.0 and <1.5 μmol/L/h, respectively) [46]. Of these 186 infants, 129 (69.4%) (infant No. 58–186) unusually carried four *IDS* gene variation combinations, i.e., [c.103+34_56dup]+[c.684A>G] + [c.851C>T]+[c.1180+184T>C]. The infants who carried these variation combinations had reduced IDS enzyme activities with an average of 5.5 (±4.8) μmol/g protein/4h compared to the reference values of 12.89–131.83 μmol/g protein/4 h. The uGAG tests were defined as being negative, with distinctly lower values of DS (0.16 ± 0.15 μg/mL) and HS (0.15 ± 0.13 μg/mL) than the cut-off values (<0.80 and 0.78 μg/mL, respectively). No MPS symptoms were found, although many of their family members, particularly from maternal relatives with the same gene variation combination, had been recalled for physiological and MPS laboratory examinations. Many hot spot mutations were verified in this study (infant No. 2–23, 24–31, and 35–40) including [c.1499C>T], [c.1478G>A], and [c.301C>T] that did not cause extraordinary IDS enzyme activities (25 ± 10.8, 23.9 ± 11.1, and 26.1 ± 9.9 μmol/g protein/4 h, respectively); meanwhile, the amounts of urinary DS and HS measured by tandem mass spectrometry were normal. Nine infants were confirmed to have MPS II (infant No. 1, 42–43, 45–46, 47, 52, 56, and 57) with nucleotide variations [c.254C>T], [c.817C>T], [c.1025A>G], [c.311A>T], [c.1400C>T], [c.1007-1666_c.1180+2113 delinsTT], and *IDS* inversion. The IDS enzyme activities were all deficient, and the values of urinary DS and HS were all significantly elevated and far higher than the cut-off values. In addition, the DMB/Cre ratio was notably elevated in most of the confirmed MPS II infants (7/9), and this was seldom observed in the infants with other types of MPS. Others infants with single missense mutations including [c.1513T>C], [c.805G>A], [c.659T>C], [c.890G>A], [c.589C>T], [c.851C>T], [c.851C>T]+[c.1180+184T>C], and [c.142C>T] had reduced (to normal) IDS enzyme activities, and negative uGAG tests, except for those with the two variations i.e., [c.659T>C] and [c.589C>T], in whom the HS quantities were slightly elevated (3.69 and 1.46 μg/mL).

### 3.3. Interpretation of the Results for MPS IVA

A total of 42 infants were referred to MMH for MPS IVA confirmation, three of whom (infant No. 1, 36, and 38) were confirmed to have nucleotide variations [c.953T>G/Heterozygote], [c.139G>A/Heterozygote], and [c.190_191delinsAT]+ [c.1108C>T]. Infant No. 1 and No. 36 only had one mutation allele, even though RNA sequencing was also performed. The GALNS enzyme activities were deficient (0.0 and 1.50 μmol/g protein/h, respectively), and significantly lower than the reference values (5.9–27.8 μmol/g protein/h). Another infant (No. 38) with two nucleotide variations [c.190_191delinsAT] and [c.1108C>T] had deficient GALNS enzyme activity of about 0.6 μmol/g protein/h. Notably, uKS quantification values for these three infants increased and higher than the cut-off value of 7.9 μg/mL. Ten infants (infant No. 6, 7, 10, 11, 19, 20, 21, 23, 26, and 32) were classified into the highly suspected group, in whom two nucleotide variations were found and identified, including [c.887C>T]+[c.953T>G], [c.857C>T]+[c.953T>G], [c.857C>T]+[c.1127G>A], [c.638C>T]+[c.953T>G], [c.857C>T]+[c.857C>T], [c.857C>T]+[c.*3C>G], [c.131G>T]+[c.985C>A], and [c.704C>A]+[c.887C>T]. The GALNS enzyme activities showed reductions ranging from 0.3 to 1.8 μmol/g protein/h, and the uKS quantification values were negative. A total of 24 infants (infant No. 2–5, 8–9, 12–18, 22, 24–25, 33, 34, 35, 37, 39, 40, 41, and 42) with one nucleotide variation were classified into the carrier group, including [c.953T>G], [c.782T>C], [c.857C>T], [c.319G>A], [c.374C>T], [c.1496C>T], [c.425A>G], [c.706C>T], [c.887C>T], [c.106_111delCTGCTC], [c.190_191delinsAT], [c.1108C>T], and [c.1019G>A]. The GALNS enzyme activities were reduced, ranging from 0.8 to 4.4 μmol/g protein/h, and the uKS quantification values were all lower than the cut-off value. Five referred infants (infant No. 27–31) were confirmed to not have MPS IVA. No nucleotide variations were found; however, their leukocyte GALNS activities were slightly reduced (3.15 (±1.0) μmol/g protein/h). The uKS quantification values were all negative, consistent with the definition of a non-MPS diagnosis.

### 3.4. Interpretation of the Results for MPS VI

Fifteen infants were referred to MMH for MPS VI confirmation. None of these infants were confirmed to have MPS VI. Three infants were classified into the highly suspected group, nine in the carrier group, and one in the non-MPS VI group. The three highly suspected MPS VI infants (infant No. 4, 7, 14) carried two nucleotide variations, i.e., [c.313-26T>C]+[c.1143-27A>C]+[c.1072G>A], [c.424A>G]+[c.1072G>A], and [c.43C>G]+[c.245T>C]. The ARSB enzyme activities were reduced to a low minimum of the normal reference value (14-228 μmol/g protein/h); notably, the uDS quantification values were negative, and the quantities of uDS were lower than the cut-off value of 7.9 μg/mL. The 10 infants (infant No. 1, 2–3, 8, 9, 10, 11, 12, 13, and 15) in the MPS VI carrier group had only one nucleotide variation, i.e., [c.1394C>G], [c.716A>G], [c.478C>T], [c.1350G>T], [c.395T>C], [c.1277A>G], [c.1197C>G], [c.1033C>T], and [c.215T>C], that caused variations in ARSB enzyme activity from a reduction to low minimum of normal reference value, and the uDS quantification values were negative as detected by tandem mass spectrometry assay. Only two infants (infant No. 5 and 6) were classified into the non-MPS VI group, none of whom had any nucleotide variations. The ARSB enzyme activities and the uKS quantification values were all normal.

## 4. Discussion

In NBS for MPS, the goals are early detection, making an early diagnosis, and providing early therapy to effectively prevent the development of irreversible clinical manifestations. Nationwide NBS programs for MPSs in Taiwan have been implemented since August 2015, and the effectiveness of these programs has been confirmed. For example, the diagnostic age has been lowered from 4.3 years to 0.2 years, allowing the possibility of providing enzyme replacement therapy (ERT) in a timely fashion and preventing severe and irreversible symptoms. MPSs have traditionally been mainly diagnosed from clinical indications, and the outcomes of subsequent therapies have generally been below expectations. However, the updated strategy of treatment for asymptomatic MPS patients is no longer proceeding as usual. Several concepts should be implemented in order to increase the screening efficiency of NBS programs for MPSs, enhance the diagnostic accuracy for MPSs, and conduct long-term intensive follow-up examinations for asymptomatic, confirmed and highly suspected MPS patients. In this study, we report the achievement and efficiency of a 6-year NBS program for MPSs in Taiwan, and clarified the important issues of an updated “gold standard” to confirm the diagnosis of MPSs.

The conventional gold standard for MPS confirmation emphasizes the deficient activity (<5% of normal) of a particular enzyme, and the identification of nucleotide variations is used to verify the results of enzyme activity assays. According to our experience, deficient enzyme activity and identified mutation alleles cannot completely confirm a diagnosis of MPS, especially if GAG-derived disaccharides are not detected in urine. Further studies are needed to investigate factors including the pathogenicity of nucleotide variations, the efficiency of enzyme catalysis of GAG degradation, and validation of the various diagnostic assays used to confirm MPS to explain the phenomenon. Nevertheless, the presence of GAG-derived disaccharides (i.e., DS, HS, or KS) in urine is a crucial factor to accurately determine the type and severity of MPS [7]. In contrast, if GAG-derived disaccharides are not detected, the diagnosis can rule out the possibility of having MPS. As shown in the tables in this study, elevated quantities of DS, HS, or KS were usually found in the confirmed cases of MPS I, MPS II, or MPS IVA, respectively.

Four infants were diagnosed with MPS I in this study. These infants included a brother and sister and twin sisters from two unrelated families, and they carried two nucleotide variations, i.e., [c.300-3C>G]+[c.1874A>C] and [c.1037T>G]+[c.1091C>T], which caused deficiency of IDUA enzyme activities. According to ACMG classification, c.300-3C>G and c.1874A>C are defined as being “Uncertain Significance” and “Likely Pathogenic”; c.1037T>G and c.1091C>T are defined as being “Pathogenic” and “Likely Pathogenic”, and both significantly affect IDUA enzyme activities [20,21]. The twin sisters seemed to be developing some symptoms, including anterior vertebral beaking and claw hands that we intend to follow up closely.

For the infants with MPS II, nine carried hemizygous variants, i.e., c.254C>T, c.817C>T, c.1025A>G, c.311A>T, c.1400C>T, c.1007-1666_c.1180+2113delinsTT, and IDS inversion, which caused deficiency of IDS enzyme activities, ranging from 0.20 to 0.83 μmol/g protein/4h. The in vitro study revealed, the percentages of IDS activity expressed in transfected COS-7 cells of the novel missense variants were 22.6% for c.254C>T, 2.2% for c.817C>T, 41.8% for c.1025A>G, 2.2% for c.311A>T, 0.0% for c.1400C>T, and these variations were classified as being “Likely Pathogenic” according to ACMG classification. Additionally, c.1007-1666_c.1180+2113delinsTT and IDS inversion have been associated with the attenuated form of MPS II [18]. All nine infants had “positive” uGAG test results, i.e., an increase in DS ranging from 1.18 to 30.77 μg/mL and an increase in HS ranging from 1.83 to 203.35 μg/mL. The abnormal elevation of GAG-derived disaccharides was not directly associated with the severity of MPS. Generally speaking, a higher HS value (impaired degradation of HS) is associated with central nervous system disorders, and a higher DS value (impaired degradation of DS) is associated with mesenchymal abnormalities [47,48]. However, it would be very difficult to predict the severity based on the quantitative values of DS and HS alone; particularly, as all of the confirmed MPS II infants were asymptomatic. Three infants received ERT accompanied with hematopoietic stem-cell transplantation (HSCT) due to a definite family history, with satisfactory outcomes, i.e., increase in leukocyte IDS enzyme activities and effective increase in the clearance of GAG accumulation of up to 80%. Two infants (infant No. 34 and 44) who carried the variants c.659T>C and c.589C>T had increased HS quantities (3.69 and 1.46 μg/mL) and mild reductions in IDS enzyme activities (10.36 and 7.80 μmol/g protein/4 h). c.659T>C is a novel variant and no ACMG classification has yet been reported. For c.589C>T, as we previously reported, the expression of IDS activity in the extracts of COS-7 cells was 74.9%, and the ACMG classification was defined as being “Likely pathogenic”. Infants with variants c.659T>C and c.589C>T had follow-up GAG examinations at 2 years and 3 months, respectively, and their HS quantities were still mildly increased (2.02 and 1.82 μg/mL), and it was very difficult to elucidate the pathogenicity in these two cases.

Three infants (infant No. 1, 36, and 38) were diagnosed with MPS IVA. Two of the three infants carried only one variant, i.e., c.953T>G in infant No. 1 and c.139G>A in infant No. 36. Another variant in these two infants was undetectable despite performing whole exome NGS for MPS genes. The other infant carried c.190_191delinsAT+ c.1108C>T, of which the variation c.190_191delinsAT was verified and showed that the transcription of exon 2 was missed by RNA sequencing analysis. Variant c.190_191delinsAT was commonly found in the confirmed MPS IVA patients and carriers; however, c.190_191delinsAT cannot be normally transcribed to produce normal protein. In RNA sequencing, the c.190_191 variant was not detectable; however, de novo RNA sequencing assembly and accompanying NGS reads showed the creation of a transcript that did not carry an entire exon 2 sequence. It is possible that the c.190_191 variant could lead to exon 2 being skipped. According to ACMG classification, c.953T>G and c.190_191delinsAT variations are defined as being “Likely Pathogenic”, c.139G>A is defined as being “Pathogenic”, and c.1108C>T is defined as being “Uncertain Significance”. The GALNS enzyme activities all showed deficiency, ranging from 0.0 to 1.5 μmol/g protein/h, and the quantification of GAG-derived disaccharide, i.e., KS, showed elevated values, ranged from 9.74 to 16.82 μg/mL. Notably, infant No. 1 carried the same variant as infants No. 2–5; however, the diagnostic consequences were quite different due to the uKS quantitative results. This is an important example showing that uGAG results important to confirm MPS. It is unclear why infant No. 1 was a confirmed MPS IVA patient, but it may be because the currently available techniques or equipment could not detect a variant. Infant No. 1 has received ERT since 7 January 2021 with satisfying outcomes observed.

Our results showed that many infants with MPS I and IVA may have more than one heterozygous or homozygous variant (autosomal recessive inheritance) and those with MPS II may have one hemizygous variant. These infants would be classified as being “highly suspected” for MPS due to the normal quantifications of DS and/or HS, or KS. For these highly suspected and confirmed infants with MPS I, MPS II, and MPS IVA, arranging long-term intensive follow-up examinations, including regular physical examinations for the earliest presenting symptoms such as otitis media, abdominal or inguinal hernia, gibbus, and coarse facial features, as well as uGAG tests are absolutely necessary.

In this study, the molecular DNA analysis revealed many hot spot nucleotide variations in the infants with Taiwanese Hunter syndrome and Morquio syndrome A. For example, a combination of four mutations (about 70%), including c.103+34_56dup, c.851C>T, c.1180+184T>C, and c.684A>G, were found in the infants with Hunter syndrome; additionally c.1499C>T, c.1478G>A, c.301C>T, and c.851C>T were found in the infants after MPS II molecular DNA analysis (11.9%, 4.32%, 3.24%, and 71.89%, respectively). According to the ACMG classification, the nucleotide variations c.1499C>T, c.1478G>A, c.301C>T, and c.851C>T are defined as being benign, likely pathogenic, benign, and uncertain significance, respectively. As previously reported, the percentages of IDS activity expressed in transfected COS-7 cells of novel variants, i.e., c.851C>T (62.3% of wild type activity), c.301C>T (97%), c.1478G>A (86.5%), c.1499C>T (77.5%). The patients with these variants did not have MPS symptoms in the present study. As reported previously, the single-nucleotide polymorphism (SNP) database of The National Center for Biotechnology Information (NCBI) showed that the variations c.1499C > T and c.1478G>A were SNPs, and we previously reported that they were not pathogenic to cause MPS II [*reference SNP: rs372205468, and rs782347729*] [6,14].

For MPS IVA, two hot spot nucleotide variations, i.e., c.953T>G and c.857C>T, were frequently found in the infants with Taiwanese Morquio syndrome A screening, (23.8% and 28.6%, respectively). According to a previous study, the nucleotide variation c.953T>G is defined as being a pathogenic gene that can cause a reduction in GALNS enzyme activity [27]. The variant c.857C>T has been reported previously [19] to cause a significant reduction in GALNS enzyme activity; however, the uKS quantification showed a normal value. According to ACMG classification, it is defined as being “Likely Pathogenic”. However, from our years of observations, this variant should be defined as being non-pathogenic rather than pathogenic.

Few studies have investigated and reported the prevalence of NBS for MPS, particularly in a large-scale feasibility study. The prevalence rates of MPS I, II, and IVA in Taiwan were 0.67, 2.92, and 4.13 per 100,000 live births, respectively. The highest birth prevalence was MPS II, accounting for 56.3% of all MPSs. MPS I, IVA, and VI accounted for 25%, 18.8%, and 0.0%, respectively. The positive predictive values of MPS I, II, and IVA NBS were 33.3% (4/12), 4.8% (9/186) and 7.1% (3/42), respectively. To largely eliminate false-positives during NBS for MPSs, a second-tier test is recommended. Stapleton et al. reported two-tier screening, and recommended that a combination of enzyme activity and multiple GAGs should be considered the gold standard for the diagnosis of MPS patients [49].

In conclusion, NBS for MPSs is an important disease screening program. Many therapies for MPS including ERT, gene therapy, HSCT and substrate reduction therapy have been developed, and their therapeutic effectiveness is remarkable if therapy begins at an early age. In this study, we proposed an update to the “gold standard” criteria required to accurately confirm a diagnosis of MPS. These updated criteria consider the relationship among nucleotide variations, leukocyte enzyme activity, and the quantity of GAG-derived disaccharides in urine. Of these three factors, uGAG-derived disaccharides is the most critical. Arranging long-term intensive follow-up examinations for asymptomatic confirmed and highly suspected MPS infants is very important to control their condition and allow the possibility of giving ERT in a timely fashion.

## Figures and Tables

**Figure 1 diagnostics-11-01583-f001:**
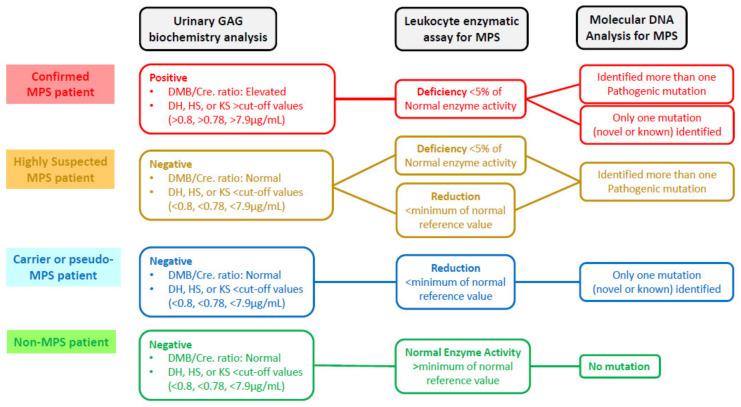
Four diagnostic groups were distinguished—the MPS confirmed group, MPS highly suspected group, MPS carrier or pseudo-MPS group, and non-MPS group—based on the results analyzed from urinary biochemistry tests, leukocyte enzyme activity assay, and molecular gene analysis.

**Figure 2 diagnostics-11-01583-f002:**
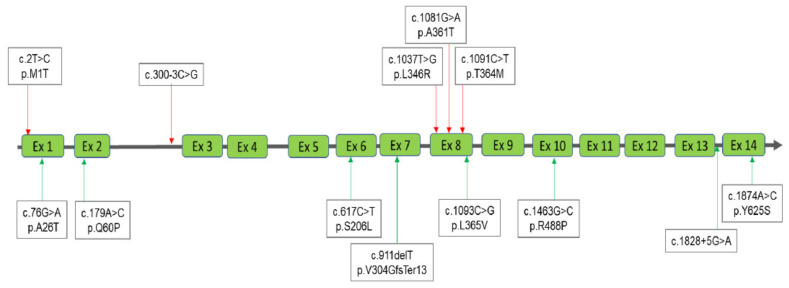
A total of 13 mutations of the *IDUA* gene underlying Taiwanese Hurler-Scheie syndrome were found; of those, five had been reported previously (red lines with arrows) and the other eight were novel mutations (green lines with arrows).

**Figure 3 diagnostics-11-01583-f003:**
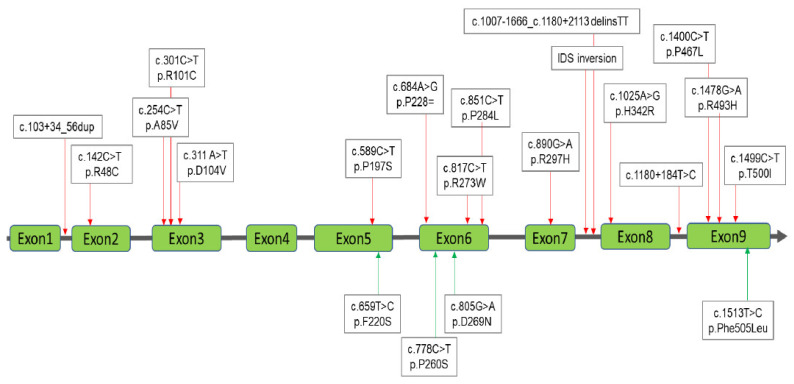
A total of 21 mutations of the *IDS* gene underlying Taiwanese Hunter syndrome were found; of those 17 had been reported previously (red lines with arrows) and the other four were novel mutations (green lines with arrows).

**Figure 4 diagnostics-11-01583-f004:**
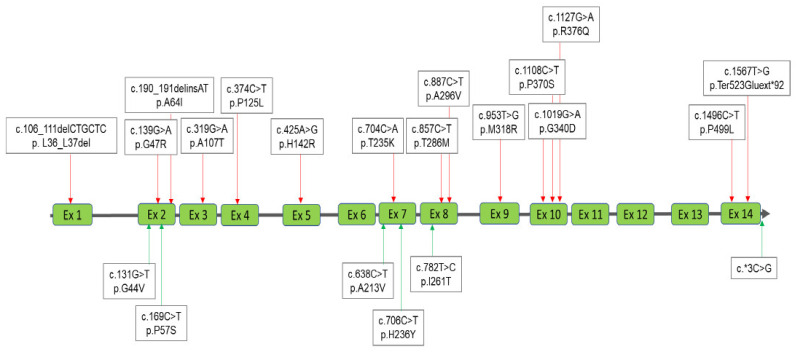
A total of 21 mutations of the *GALNS* gene underlying Taiwanese Morquio A syndrome were found; of those 15 had been reported previously (red lines with arrows) and the other six were novel mutations (green lines with arrows).

**Figure 5 diagnostics-11-01583-f005:**
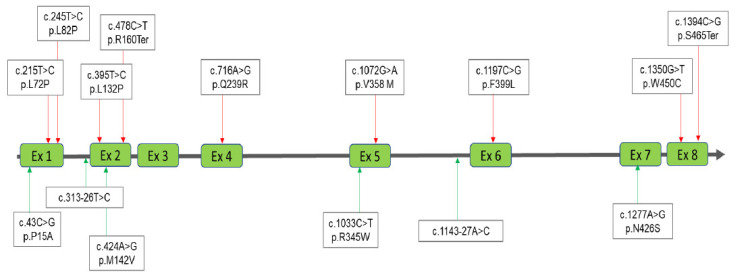
A total of 15 mutations of the *ARSB* gene underlying Taiwanese Maroteaux-Lamy syndrome were found; of those nine had been reported previously (red lines with arrows) and the other six were novel mutations (green lines with arrows).

**Table 1 diagnostics-11-01583-t001:** Biochemistry GAG tests, enzyme activity, and mutations of alpha-L-iduronidase (IDUA) gene in suspected MPSI infants.

Infant No.	Sex	DMB/Cre. Ratio	DS	HS	KS	Enzyme Activity	Nucleotide Variations	Protein Variations	ACMG Classification
1	Female	37.86	0.27	0.10	5.69	1.20	[c.343G/A]+ [c.2T/C] [19]	[p.D115N]+[p.M1T]	Uncertain SignificanceLP(VUS with minor Pathogenic evidence)+ Pathogenic
2	Male	34.02	0.11	0.17	0.53	1.50	[WT/c.1395delC]+ [WT/c.1081G>A] [20]+ [WT/c.1816G>T]	[p.G466AfsTer59]+ [p.A361T]+[p.V606L]	Pathogenic + Benign + Uncertain SignificanceLP(VUS with minor Pathogenic evidence)
3	Male	24.99	0.11	0.17	0.83	2.10	[WT/c.355G>T]+ [WT/c.617C>T]	[p.D119Y]+[p.S206L]	Uncertain SignificanceLP(VUS with minor Pathogenic evidence) + Uncertain SignificanceLP(VUS with minor Pathogenic evidence)
4	Male	24.89	0.05	0.29	0.16	3.88	[WT/c.179A>C]	[p.Q60P]	Uncertain SignificanceLP(VUS with minor Pathogenic evidence)
5	Female	36.34	0.01	0.04	2.57	3.82	[−]	[−]	-
6	Male	30.99	0.04	0.06	0.10	9.62	[−]	[−]	-
7	Female	26.82	0.20	0.01	6.49	21.60	[c.76G>A]+ [c.911delT]	[p.A26T ]+ [p.V304Gfs*13]	Uncertain SignificanceLP(VUS with minor Pathogenic evidence) + Pathogenic
8	Male	17.97	21.18	11.34	2.0	1.60	[c.300-3C>G] [21]+ [c.1874A>C]	[−]+[p.Y625S]	Uncertain Significance + Likely Pathogenic
9	Female	33.61	5.91	9.38	0.28	0.46	[c.300-3C>G] [21]+ [c.1874A>C]	[−]+[p.Y625S]	Uncertain Significance + Likely Pathogenic
10	Female	204.70	176.96	28.51	1.78	0.70	[c.1037T>G] [21]+ [c.1091C>T] [22]	[p.L346R]+[p.364M]	Pathogenic + Likely Pathogenic
11	Female	239.25	296.10	33.55	1.37	0.80	[c.1037T>G] [21]+ [c.1091C>T] [22]	[p.L346R]+[p.364M]	Pathogenic + Likely Pathogenic
12	Male	18.53	0.09	0.17	0.78	0.95	[WT/c.1093C>G]+ [WT/c.1463G>C]+ [WT/c.1828+5G>A]	[p.L365V]+[p.R488P]+ [−]	Uncertain Significance + Uncertain Significance + Uncertain Significance

IDUA enzyme activity (Ref. 5.9–27.8 μmol/g protein/h); GAG quantification (DMB/Cre ratio) and the normal reference values based on the age groups: <6 mos: <70.68 mg /mmol cre.; 6 mos–2 yrs: <46.80 mg /mmol cre.; 2–17 yrs: <20.98 mg /mmol cre.; and >18 yrs: <12.62 mg /mmol cre. Quantitative analyses of GAG-derived disaccharides (DS, S, or KS) by tandem mass spectrometry assay. The normal cut-off values: DS < 0.80 μg/mL; HS < 0.78 μg/mL; and KS < 7.90 μg/mL.

**Table 2 diagnostics-11-01583-t002:** Biochemistry GAG tests, enzyme activity, and mutations of Iduronate 2-Sulfatase (IDS) gene in suspected MPSII infants.

Infant No.	Sex	DMB/Cre. Ratio	DS	HS	KS	Enzyme Activity	Nucleotide Variations	Protein Variations	ACMG Classification
1	Male	78.58	11.59	12.36	0.25	0.83	[c.254C>T] [18]	[p.A85V]	Likely Pathogenic
2-23	Male	30.3 ± 10.8	0.09 ± 0.06	0.1 ± 0.09	0.36 ± 0.26	25 ± 10.8	[c.1499C>T] [18]	[p.T500I]	Benign
24–31	Male	29 ± 11.1	0.16 ± 0.1	0.06 ± 0.04	0.79 ± 0.6	23.9 ± 11.1	[c.1478G>A] [18]	[p.R493H]	
32	Male	27.69	0.18	0.2	0.08	5.93	[c.1513T>C]	[p.F505L]	Likely Pathogenic
33	Male	46.09	0.17	0.53	1.32	17.68	[c.805G>A]	[p.D269N]	
34	Male	73.20	0.21	3.69	4.03	10.36	[c.659T>C]	[p.F220S]	
35–40	Male	18.1 ± 6.4	0.06 ± 0.04	0.14 ± 0.04	4.92 ± 1.9	26.1 ± 9.9	[c.301C>T] [23]	[p.R101C]	Benign
41	Male	69.67	0.08	0.04	3.47	9.20	[c.890G>A] [18]	[p.R297H]	Likely Pathogenic
42–43	Male	77.3/16.73	7.39/5.48	1.83/24.9	6.13/0.58	0.20/0.41	[c.817C>T] [18]	[p.R273W]	Likely Pathogenic
44	Male	63.82	0.38	1.46	6.35	7.80	[c.589C>T] [18]	[p.P197S]	Likely Pathogenic
45–46	Male	59.46/70.9	1.18/21.21	8.22/12.06	1.49/6.47	1.26/0.40	[c.1025A>G] [18]	[p.H342R]	Likely Pathogenic
47	Male	113.95	15.62	103.44	1.42	0.32	[c.311A>T] [18]	[p.D104V]	Likely Pathogenic
48–51	Male	27.9 ± 5.5	0.03	0.12 ± 0.05	0.22 ± 0.02	8.7 ± 8	[c.851C>T] [18]	[p.P284L]	Uncertain Significance
52	Male	153.16	21.4	30.01	0.11	0.27	[c.1400C>T] [18]	[p.P467L]	Likely Pathogenic
53	Male	55.87	0.14	0.38	0.32	2.93	[c.851C>T] [18]+ [c.1180+184T>C] [18]	[p.P284L]+[−]	Uncertain Significance + [−]
54–55	Male	38.73/26.2	0.01/0.08	0.75/0.38	0.03/0.59	16.27/23.78	[c.142C>T] [18]	[p.R48C]	Likely Pathogenic
56	Male	177.96	30.77	203.35	0.31	0.99	[c.1007-1666_c.1180+2113 delinsTT] [18]		Pathogenic
57	Male	104.84	11.93	175.36	0.15	0.13	IDS inversion [24,25]		
58–186	Male	29.7 ± 12.5	0.16 ± 0.15	0.15 ± 0.13	0.43 ± 0.29	5.5 ± 4.8	[c.103+34_56dup] [18]+ [c.684A>G] [18]+ [c.851C>T] [18]+ [c.1180+184T>C] [18]	[−]+[p.Pro228=]+ [p.P284L] +[−]	Uncertain Significance + Benign + Uncertain Significance + [−]

IDS enzyme activity (Ref. 12.89~131.83 μmol/g protein/4 h); GAG quantification (DMB/Cre ratio) and the normal reference values based on the age groups: <6 mos: <70.68 mg /mmol cre.; 6 mos-2 yrs: <46.80 mg /mmol cre.; 2–17 yrs: <20.98 mg /mmol cre.; and >18 yrs: <12.62 mg /mmol cre. Quantitative analyses of GAG-derived disaccharides (DS, S, or KS) by tandem mass spectrometry assay. The normal cut-off values: DS < 0.80 μg/mL; HS < 0.78 μg/mL; and KS < 7.90 μg/mL.

**Table 3 diagnostics-11-01583-t003:** Biochemistry tests, enzyme activity, and mutations of Galactosamine (N-acetyl)-6-sulfatase (GALNS) gene in suspected MPS IVA infants.

Infant No.	Sex	DMB/Cre. Ratio	DS	HS	KS	Enzyme Activity	Nucleotide Variations	Protein Variations	ACMG Classification
1	Male	86.81	0.12	0.01	10.64	0.00	[WT/c.953T>G] [26]	[p.M318R]	Likely Pathogenic
2–5	Male	33.6 ± 7.8	0.17 ± 0.1	0.09 ± 0.07	2.01 ± 1.5	1.63 ± 1.4	[WT/c.953T>G] [26]	[p.M318R]	Likely Pathogenic
6–7	Male, Female	33.37 ± 11.7	0.06 ± 0.01	0.41 ± 0.5	0.81 ± 0.2	1.75 ± 0.07	[WT/c.887C>T] [27]+ [WT/c.953T>G] [26]	[p.A296V]+ [p.M318R]	Uncertain Significance + Likely Pathogenic
8–9	Male	39.61 ± 0.2	0.12 ± 0.05	0.03 ± 0.01	0.6 ± 0.4	3.0 ± 0.1	[WT/c.782T>C]	[p.I261T]	Likely Pathogenic
10–11	Female	33.67 ± 3.9	0.19 ± 0.1	0.02 ± 0.01	0.56 ± 0.06	0.5 ± 0.3	[WT/c.857C>T] [27]+ [WT/c.953T>G] [26]	[p.T286M]+ [p.M318R]	Likely Pathogenic + Likely Pathogenic
12–18	3 Male, 4 Female	29.9 ± 5.9	0.27 ± 0.08	0.06 ± 0.05	0.97 ± 0.7	2.0 ± 0.4	[WT/c.857C>T] [27]	[p.T286M]	Likely Pathogenic
19	Male	48.17	0.02	0.05	3.54	1.40	[WT/c.857C>T] [27]+ [WT/c.1127G>A] [28,29,30]	[p.T286M]+ [p.R376Q]	Likely Pathogenic + Likely Pathogenic
20	Female	32.67	0.64	0.04	0.71	0.40	[WT/c.638C>T]+ [WT/c.953T>G] [26]	[p.A213V]+ [p.M318R]	Likely Pathogenic + Likely Pathogenic
21	Male	38.14	0.17	0.02	0.06	1.50	[c.857C>T] [27]	[p.T286M]	Likely Pathogenic
22	Female	34.21	0.13	0.36	8.18	1.70	[WT/c.319G>A] [31]	[p.A107T]	Likely Pathogenic
23	Male	15.32	0.04	0.46	0.87	2.40	[WT/c.857C>T] [27]+ [c.*3C>G]	[p.T286M]+[−]	Likely Pathogenic + Uncertain Significance
24–25	Female	17.43 ± 0.1	0.24 ± 0.1	0.21 ± 0.1	0.78 ± 0.03	1.55 ± 1.0	[WT/c.374C>T] [32]	[p.P125L]	Likely Pathogenic
26	Male	19.02	0.03	0.16	0.90	0.80	[WT/c.131G>T]+ [WT/c.985C>A]	[p.G44V]+ [p.H329N]	Likely Pathogenic + Likely Pathogenic
27–31	Male	24.8 ± 7.3	0.32 ± 0.2	0.07 ± 0.03	0.41 ± 0.3	3.15 ± 1.0	[−]	[−]	-
32	Male	41.54	0.07	0.28	0.67	3.50	[WT/c.704C>A] [33]+ [WT/c.887C>T] [27]	[p.T235K]+ [p.A296V]	Likely Pathogenic + Uncertain Significance
33	Female	21.12	0.08	0.17	0.44	4.10	[WT/c.1496C>T] [34]	[p.P499L]	Likely Pathogenic
34	Female	56.83	0.11	0.05	0.23	3.80	[WT/c.425A>G] [35]	[p.H142R]	Likely Pathogenic
35	Female	33.87	0.14	0.16	0.43	1.40	[WT/c.706C>T]	[p.H236Y]	Likely Pathogenic
36	Male	49.59	0.04	0.01	16.82	1.50	[WT/c.139G>A] [36,37]	[p.G47R]	Pathogenic
37	Male	18.45	0.04	0.03	5.15	3.60	[WT/c.887C>T] [27]	[p.A296V]	Uncertain Significance
38	Female	33.45	0.04	0.07	9.74	0.60	[WT/c.190_191delinsAT] [27]+[WT/c.1108C>T] [27]	[p.A64I]+ [p.P370S]	Likely Pathogenic + Uncertain Significance
39	Male	50.38	0.05	0.09	0.59	2.10	[WT/c.106_111delCTGCTC] [38]	[p.L36_L37del]	Likely Pathogenic
40	Male	7.80	0.02	0.03	4.80	2.00	[WT/c.190_191delinsAT] [27]	[p.A64I]	Uncertain Significance
41	Male	32.38	0.03	0.08	1.10	2.10	[WT/c.1108C>T] [27]	[p.P370S]	Uncertain Significance
42	Female	12.76	0.02	0.23	5.08	4.40	[WT/c.1019G>A] [35,36]	[p.G340D]	Pathogenic

GALNS enzyme activity (Ref. 5.9–27.8 μmol/g protein/h); GAG quantification (DMB/Cre ratio) and the normal reference values based on the age groups: <6 mos: <70.68 mg /mmol cre.; 6 mos-2 yrs: <46.80 mg /mmol cre.; 2-17 yrs: <20.98 mg /mmol cre.; and >18 yrs: <12.62 mg /mmol cre. Quantitative analyses of GAG-derived disaccharides (DS, S, or KS) by tandem mass spectrometry assay. The normal cut-off values: DS < 0.80 μg/mL; HS < 0.78 μg/mL; and KS < 7.90 μg/mL.

**Table 4 diagnostics-11-01583-t004:** Biochemistry tests, enzyme activity, and mutations of Arylsulfatase B (ARSB) gene in suspected MPS VI infants.

Infant No.	Sex	DMB/Cre. Ratio	DS	HS	KS	Enzyme Activity	Nucleotide Variations	Protein Variations	ACMG Classification
1	Male	44.41	0.49	0.10	0.68	16.30	[WT/c.1394C>G] [39]	[p.S465Ter]	Pathogenic
2-3	Female	22.3 ± 19	0.06 ± 0.04	0.06 ± 0.05	2.5 ± 1.4	13.7 ± 8.5	[WT/c.716A>G] [39]	[p.Q239R]	Likely Pathogenic
4	Female	45.17	0.03	0.01	0.08	5.00	[WT/c.313-26T>C]+ [WT/c.1143-27A>C]+ [WT/c.1072G>A] [40]	[−]+[−]+[p.V358 M]	Benign + Benign + Benign
5–6	Male	38.2 ± 4	0.33 ± 0.4	0.03 ± 0.01	0.29 ± 0.07	18.5 ± 11.3	[−]	[−]	-
7	Male	21.48	0.07	0.11	0.17	13.60	[WT/c.424A>G]+ [c.1072G>A] [40]	[p.M142V]+ [p.V358M]	Likely Pathogenic + Benign
8	Male	37.54	0.26	0.21	1.65	1.10	[WT/c.478C>T] [41]	[p.R160Ter]	Pathogenic
9	Male	14.30	0.10	0.15	0.16	23.00	[WT/c.1350G>T] [42]	[p.W450C]	Pathogenic
10	Female	26.97	0.15	0.09	1.22	20.80	[WT/c.395T>C] [39]	[p.L132P]	Likely Pathogenic
11	Male	5.50	0.58	0.04	0.87	11.00	[WT/c.1277A>G]	[p.N426S]	Likely Pathogenic
12	Male	8.49	0.19	0.11	0.37	23.00	[WT/c.1197C>G] [43]	[ p.F399L ]	Pathogenic
13	Female	28.02	0.36	0.11	0.85	7.50	[WT/c.1033C>T]	[p.R345W]	Likely Pathogenic
14	Female	29.59	0.34	0.31	0.75	17.50	[WT/c.43C>G]+ [WT/c.245T>C] [44]	[p.P15A]+[p.L82P]	Uncertain Significance LP(VUS with minor Pathogenic evidence) + Likely Pathogenic
15	Male	26.23	0.03	0.36	0.98	5.00	[WT/c.215T>C] [45]	[p.L72P]	Likely Pathogenic
1	Male	44.41	0.49	0.10	0.68	16.30	[WT/c.1394C>G] [39]	[p.S465Ter]	Pathogenic
2–3	Female	22.3 ± 19	0.06 ± 0.04	0.06 ± 0.05	2.5 ± 1.4	13.7 ± 8.5	[WT/c.716A>G] [39]	[p.Q239R]	Likely Pathogenic

ARSB enzyme activity (Ref. 14–228 μmol/gm protein/h); GAG quantification (DMB/Cre ratio) and the normal reference values based on the age groups: <6 mos: <70.68 mg /mmol cre.; 6 mos-2 yrs: <46.80 mg /mmol cre.; 2-17 yrs: <20.98 mg /mmol cre.; and >18 yrs: <12.62 mg /mmol cre. Quantitative analyses of GAG-derived disaccharides (DS, S, or KS) by tandem mass spectrometry assay. The normal cut-off values: DS < 0.80 μg/mL; HS < 0.78 μg/mL; and KS < 7.90 μg/mL.

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
