# Peer review of "Nationwide Newborn Screening Program for Mucopolysaccharidoses in Taiwan and an Update of the “Gold Standard” Criteria Required to Make a Confirmatory Diagnosis"

_diagnostics, 2021, doi:10.3390/diagnostics11091583_

Round 1

Reviewer 1 Report

The paper by Chung et al. describes the result of a NBS for 4 treatable MPS performed in China and it is indicating a way to detect the affected newborns by linking mass spec GAG disaccharide determination, enzyme activity and mutation analysis. The paper might be of interest for the readers, the casistic is relevant and the indications given might be useful.

However, I think that the Authors should comment about the following issues:

1) please indicate clearly which method you used for initial NBS, genetic analysis or GAG determination? It is not clear,  please give some detail on it, and the reason of choice (cost, sensitivity…)

2) it would be interesting to know whether all the patients received therapy or not and which therapy they received I.e ERT, HSCT, or combination? This is said only for 3 MPSII patient, we miss information for all  the other patients.

Author Response

Responses to the reviewer’s comments.

We deeply appreciate the reviewer’s kind and affirmative comment and we provide our point-by-point responses to the notes of the reviewer.

Reviewer 1.

  • please indicate clearly which method you used for initial NBS, genetic analysis or GAG determination? It is not clear,  please give some detail on it, and the reason of choice (cost, sensitivity…)

Response:           We had added more details in the section of MATERIALS AND METHODS according to the reviewer’s comments. Please see the section of MATERIALS AND METHODS on 4.1. NBS for MPSs and samples referred (page No. 8); 4.3.1. Total GAG quantification (page No. 8); and 4.3.2. GAG-derived disaccharide quantification by tandem mass spectrometry assay (page No.9).

The reason of choice for the methods we were applied is based on the following points, including 1. All the methods are the most extensive protocols which are conventionally performed for MPS screening and confirmatory diagnoses in the world; 2. All the methods are all with highly specific and with highly sensitive that make the MPS diagnosis more feasible and more reliable; and 3. The cost of the experiments is acceptable and affordable.              

  • it would be interesting to know whether all the patients received therapy or not and which therapy they received I.e ERT, HSCT, or combination? This is said only for 3 MPSII patient, we miss information for all the other patients.

Response:           Three of the nine infants have a distinct family history, and these three infants have received ERT accompanied by HSCT. As we mentioned in the text, the outcomes after ERT and HSCT are satisfactory, i.e. leukocyte IDS enzyme activities increased and effective clearance of GAG accumulation. In addition, One confirmed MPS IVA infant (infant No. 1 with c.953T>G) has received ERT since January 7, 2021 due to undetectable leukocyte GALNS enzyme activity and increase of urinary KS quantification (Please see the section of DISCUSSIONS AND CONCLUSIONS on page No. 6). Four infants included a brother and sister and twin sisters from two unrelated families were diagnosed with MPS I; however the twin sisters’ parents are not willing to receive ERT even though few mild manifestations are observed. For the other infants defined as being at risk of having MPS, long-term follow up is obviously necessary and has been arranged in 6-month intervals for regular physical and biochemistry examinations.

Reviewer 2 Report

This paper brings together the most important elements in the screening for LSD highlighting the need for confirmatory diagnosis based on a combination of variant analysis, urinary GAG determination and enzyme activity. In a reported screened population of ~600K babies present very strong outcome data on screening for these LSD, and presents high value to those contemplating introducing the screening for LSD. Whilst there may be ethnic variations in the variants identified by this group it nevertheless the value of the information is of high significance.

Author Response

Reviewer 2.

This paper brings together the most important elements in the screening for LSD highlighting the need for confirmatory diagnosis based on a combination of variant analysis, urinary GAG determination and enzyme activity. In a reported screened population of ~600K babies present very strong outcome data on screening for these LSD, and presents high value to those contemplating introducing the screening for LSD. Whilst there may be ethnic variations in the variants identified by this group it nevertheless the value of the information is of high significance.

Response:         We deeply appreciate the reviewer’s kind and affirmative comment. In our study, we indeed found some hot spot variations underline Taiwanese population, as the reviewer’s comment, there may be ethnic variations in the variants identified by Taiwanese group it nevertheless the value of the information is of high significance.